# Temporomandibular Disorders and Bruxism among Sex Workers—A Cross Sectional Study

**DOI:** 10.3390/jcm11226622

**Published:** 2022-11-08

**Authors:** Ilana Eli, Adi Zigler-Garburg, Efraim Winocur, Pessia Friedman-Rubin, Tamar Shalev-Antsel, Shifra Levartovsky, Alona Emodi-Perlman

**Affiliations:** Department of Oral Rehabilitation, The Maurice and Gabriela Goldschleger School of Dental Medicine, Sackler Faculty of Medicine, Tel Aviv University, Tel Aviv 6139001, Israel

**Keywords:** temporomandibular disorders (TMD), bruxism, sex workers, DC/TMD

## Abstract

Sex workers are a highly underprivileged population which is present all around the world. Sex work is associated with negative social stigma which affects all aspects of the sex workers’ lives including healthcare, service providers and police. The stigma may result in increased stress, mental health problems, feelings of isolation and social exclusion. In the present study, 36 sex workers (SW) and 304 subjects from the general population in Israel (GP) were evaluated for the presence of bruxism and Temporomandibular disorders (TMD), with the use of Diagnostic Criteria for Temporomandibular Disorders (DC/TMD- Axis I). When compared to the general population, sex workers presented larger maximal assisted mouth opening and higher prevalence of the following TMD diagnoses: Disc displacement with reduction, Myalgia, Myofascial pain with referral, Arthralgia (left and right) and Headache attributed to TMD. The odds of sex workers suffering from one of these diagnoses were twice to five times higher than those of the general population. The study shows that health problems of sex workers go beyond venereal diseases, HIV and mental disorders which are commonly studied. Oral health, TMD and oral parafunctions are some of the additional health issues that should be addressed and explored in this population.

## 1. Introduction

Temporomandibular disorders (TMD) are defined as a group of musculoskeletal and neuromuscular conditions that involve temporomandibular joints, masticatory muscles and all associated structures of mastication and associated tissues [1]. The most common features of TMD are regional pain, limited jaw movements and acoustic sounds from the temporomandibular joints (TMJ) during motion [2]. The prevalence of TMD in the general population is about 10–15% [3,4,5], affecting mainly young and middle-aged women.

The etiology of TMD is complex and multifactorial, involving genetic predisposition, trauma, hypermobility, physiological and psychosocial factors (i.e., anxiety, stress, depression, coping strategies, catastrophizing), all of which are capable of influencing the onset of pain and lead to persistent orofacial pain [1,6,7,8]. A Prospective Evaluation and Risk Assessment (OPPERA) study found that TMD onset was strongly associated with somatic symptoms, previous life events, perceived stress and negative affect [9].

Bruxism is considered a repetitive jaw muscle activity characterized by clenching or grinding of the teeth and/or bracing or thrusting of the mandible [10]. In otherwise healthy individuals bruxism should not be considered as a disorder but rather a behavior that may be a risk (and/or protective) factor for certain clinical consequences [11]. Bruxism is further separated into two distinct muscle behaviors according to whether they appear during wakefulness (Awake bruxism—AB) or during sleep (Sleep bruxism—SB). The prevalence of AB in the general population is about 22–30%. The prevalence of SB is about 16% among young adults and 3–8% among adults [12]. It is now thought that while AB is associated with anxiety and stress sensitivity [13,14,15,16], SB is considered to be more central in origin [17]. 

Stress is one of the psychological factors involved in TMD pathophysiology. Increased levels of stress in patients with TMD are associated with elevated levels of cortisol, hyperactivity of the Hypothalamus-Pituitary-Adrenal axis and increased bio-electric activity of the masticatory muscles [18].

Additional origins of TMD pain can be due to voluntary muscle contraction. In contrast to cutaneous pain, which is comparatively well understood, the molecular mechanisms for muscle pain are not fully understood. Muscle pain and fatigue can lead to syndromes such as chronic fatigue syndrome (CFS) and fibromyalgia which are frequently associated with a variety of other syndromes, such as TMD [19].

Several studies have been published regarding TMD and bruxism in various populations [20,21]. One of the populations which has not been studied with regard to TMD and bruxism are sex workers, namely- subjects engaged in prostitution [22]. 

The exact prevalence of commercial sex workers is debated, varies according to geographical location and legal status and is estimated as ranging between 0.7–7.4% [23]. Women make up the majority of the sex work population, with estimated proportion of around 85–90% [24].

Sex work is associated with negative social stigma which affects all aspects of the sex workers’ lives including healthcare, service providers and police [25]. The stigma may result in increased stress, mental health problems, feelings of isolation and social exclusion [26]. Sex workers (SW) tend to experience high levels of violence which in turn may lead to high rates of anxiety, stress and post-traumatic stress disorder [27,28]. All of the above may influence the onset of both TMD and bruxism. 

In addition, substance abuse such as smoking, drugs and/or alcohol is common among this population. A comprehensive cross-sectional study on female SW showed that about 72% drink alcohol, with many consuming very large quantities. About 20–30% of the female SW use drugs and/or alcohol to help them cope with being SW [28]. Long-term drug abuse leads to a high prevalence of oral motor behavior and signs/symptoms of TMD [29], while smoking affects the painful perception of TMD patients and influences co-morbid aspects such as fatigue worsening, pain control, sleep quality and psychological distress [30]. Therefore, substance abuse may act as a risk factor for SB and TMD.

As part of increasing social involvement, a project entitled “The tooth fairy” was initiated in the Tel Aviv University Dental School in the year 2016. The project aims to supply basic dental care, free of charge, to underprivileged populations such as sex workers. Sex workers were invited to arrive for dental treatment through municipal social workers involved with these populations.

The aim of our study was to evaluate the extent of TMD and bruxism among sex workers in Israel as compared to the general population.

## 2. Materials and Methods

The study received the approval of the Tel Aviv University Ethical Committee (No. 0001536-2).

### 2.1. Population

36 sex workers (SW), who arrived for treatment at the Dental School in the years 2018–2022, gave their consent to participate in the study. For comparison 304 subjects from the general population in Israel (GP), who arrived for dental treatment in the same time period, were also included in the analyses. All subjects supplied a written informed consent for participation.

Inclusion criteria were a minimal age of 18 years and a complete anamnesis, including a full clinical examination according to the Diagnostic Criteria for Temporomandibular Disorders (DC/TMD) Axis I (http://rdc-tmdinternational.or) Accessed on 2 January 2018. Since the reliability of the DC/TMD has been tested only on adult populations, patients under 18 years were excluded. As some medical conditions and/or drugs can affect TMD and/or bruxism diagnosis, additional exclusion criteria were patients suffering from uncontrolled hormonal disease, neurological disturbances, psychiatric problems, neoplasm, history of facial or cervical injury, patients using anti-depressants and/or myo-relaxants and substance abusers. 

Of the GP group, 6 subjects were excluded due to psychiatric conditions, 5 due to drug abuse and 5 due to neurological conditions. Of the SW group, 1 subject was excluded due to an unbalanced hormonal condition. 

Final study population included a total of 323 patients as follows:

GP group: No. = 288, 52.4% male, mean age 36.2 ± 13.4 years.

SW group: No. = 35, 60% male, mean age 30.0 ± 8.3 years. 

SW were younger (*p* < 0.001, *t*-test). No differences were found between groups gender wise. 

### 2.2. Data Collection

Anamnesis, clinical examination and TMD diagnosis were carried out according to the DC/TMD (Axis I). All procedures were carried out by senior dental students, under a close supervision and verification of senior faculty members who have been calibrated and certified in the official DC/TMD Training and Calibration Course at the Department of Orofacial Pain and Jaw function at the Faculty of Odontology at Malmö University, Sweden. 

1. **TMD Diagnosis**: TMD diagnosis was based on the criteria of the international DC/TMD consortium (http://rdc-tmdinternational.or) accessed on 25 January 2018. In order to obtain an Axis I TMD diagnosis, all patients were requested to complete the TMD pain screener (official Hebrew version) which is part of the DC/TMD Axis I self-report instruments [31]. The TMD pain screener is a six-item questionnaire. The maximal score is 7, with score of >3 indicating the presence of TMD [32]. 

Pain screener score, in addition to Axis I clinical evaluation, enabled the following diagnoses:(i)Pain related TMD: myalgia, myofascial pain with referral, arthralgia (right or left), headache attributed to TMD.(ii)Articular TMD disorders: Disc displacement with reduction (DDwR), Disc displacement with reduction with intermittent locking (DDwRwIL), Disc displacement without reduction with limited opening (DdwoRwLO), Disc displacement without reduction without limited opening (DDwoRwoLO), Degenerative joint disease (DJD), Subluxation (either right or left for each diagnosis).

DC/TMD Axis II questionnaires were avoided due to the built-in social sensitivity of subjects belonging to the SW group.

2. **Probable Sleep & Probable Awake Bruxism (pAB & pSB):** Diagnosis of pSB and/or pAB was carried out according to the grading suggested by the international consensus on the assessment of bruxism [11], namely, when a respondent reported awareness of performing bruxing behavior according to the Oral Behavior Checklist [31], in addition to clinical signs supporting such behavior as follows: (i)Probable SB (pSB)—subject reported awareness of preforming bruxing muscle activity during the past 30 days, for at least 1–3 nights/week [31] and presented at least one clinical sign supporting such a behavior such as presence of Linea Alba and/or tongue scalloping and/or dental wear [33].(ii)Probable AB (pAB)– subject reported awareness of preforming bruxing muscle activity, most of the time, during waking hours, during the past 30 days and presented at least one clinical sign supporting such a behavior such as presence of Linea Alba and/or tongue scalloping and/or dental tooth wear [33].

3. **Report of oral parafunctional activity:** Subjects were requested to report performing the following oral behaviors, during the past 30 days: nail biting; chewing gum; holding or biting hard objects between the teeth (pens, pencils, hair, etc.); biting soft oral tissues. The report was given on a 5 points Likert scale as follows: none of the time, a little of the time, some of the time, most of the time, all of the time. Patients who reported performing at least one parafunctional behavior, most of the time, were recorded as positive for parafunctional activity.

4. **Final analyses referred to the following variables:**
TMD pain according to TMD pain screener [31].Performing oral parafunctional activities, as defined above (yes/no).Mouth opening, lateral and protrusive mandibular movements (in millimeters).TMD diagnosis according to Axis I of the DC/TMD.Probable SB and probable AB, as defined above (yes/no).

Statistics: Data were analyzed using SPSS software (IBM SPSS statistics 27.0, Armonk, NY, USA). Chi-square and *t*-test analyses were used to compare between groups. Risk estimate was used to determine the effect of group (GP versus SW) on the odds of bruxism and TMD diagnoses. 

## 3. Results

Groups did not differ in the use of the following medications: Selective Serotonin Reuptake Inhibitors (SSRI); Serotonin-Norepinephrine Reuptake Inhibitors (SNRI); methylphenidate (Ritalin), Neuroleptics, Analgesics, or Anxiolytics, nor in the report of medical conditions such as diabetes, hypertension, cardiovascular disease, gastroesophageal reflux disease (GERD) or other medical conditions. Since the above-mentioned medications and/or medical conditions have a potential to affect bruxism and TMD [34], the finding that there were no differences between groups allowed carrying out the analyses without introducing them as covariates. 

1.TMD pain according to TMD pain screener: 

There was a significant difference between groups in their report of TMD pain according to the TMD pain screener (24.2 among the GP group versus 48.6 in the SW group, *p* < 0.01, chi square).

There was a significant difference between groups in the percentage of subjects presenting at least one TMD symptom (46.5% in the GP group versus 71.1% in the SW group, *p* < 0.01).

2.Performing oral parafunctional activities: 

Except for soft tissue biting, there were no differences between groups in the performance of oral parafunctions (Table 1).

3.Mouth opening, lateral and protrusive mandibular movements:

Range of maximal mouth opening (unassisted and assisted), lateral and protrusive mandibular movements of the two groups is presented in Table 2. The only difference between groups was in maximal assisted opening. 

4.TMD diagnosis:

There was a significant difference between groups in the percentage of subjects presenting at least one TMD diagnosis (56.9% in the GP group versus 80.1% in the SW group, *p* < 0.01). 

TMD diagnoses of both groups are presented in Table 3. Significant differences between groups could be observed in pain disorders (myofascial pain, disc displacement with reduction, right and left arthralgia and headache associated with TMD, in addition to Disk displacement with reduction.

Chi square risk estimates of the TMD diagnoses which showed significant differences between groups are presented in Table 4.

5.Probable SB and probable AB: 

No differences between groups were found in the presence of either pAB and/or pSB, nor in the presence of clinical signs supporting pSB and/or pAB behavior (Linea Alba, abfractions, indentations on tongue and/or cheek or amount of dental wear; Table 5).

## 4. Discussion

Sex workers are a highly underprivileged population which is present all around the world. Female sex workers comprise 0.4–4.3% if the population in sub-Saharan Africa, 0.2–2.6% in Asia, 0.1–1.5% in the ex-Russian Federation, 0.4–1.4% in East Europe, 0.1–1.4% in West Europe and 0.2–7.4% in Latin America [23]. This huge population suffers (among other problems) from poor mental health and high incidence of depression and post-traumatic stress disorder [28]. Regretfully, as a result of the stigma experienced by sex workers, health and social care supplied to this group are not always adequate and/or satisfactory.

The prevalence of TMD in the general population is about 10–15%, with more than 50% presenting signs of TMDs [3,4,5,35]. In the present study, almost 57% of the general population presented at least one TMD diagnosis, which is in accord with other studies that used the DC/TMD Axis I as screening tool [36,37,38,39,40]. Among sex workers participating in the present study, the prevalence of at least one TMD diagnosis was significantly higher (80%). 

Various occupations and/or behaviors exhibit higher risk for development of TMD due to functional and or psychosocial factors that overload the stomatognathic system. For example, high-tech workers, dentists, scuba divers, instrument players in general and violin players in particular, show a high prevalence of reported TMD symptoms which indicate an increased risk for developing TMD [21,41,42,43,44,45]. 

Under-monitoring and under-treatment of sex workers [46] may present both functional and psychosocial risk factors related to the nature of their occupation. Sex work is considered as one of the most dangerous and stressful occupations [47]. A 3-year mental health survey conducted in Vancouver, Canada, found that almost 50% of sex workers were diagnosed with mental health issues, mainly depression and anxiety. A vast majority of the diagnosed subjects (95%) experienced physical and/or sexual trauma [46], which contributed further to the development of post-traumatic stress syndrome (PTSD) [48,49]. Patients with severe PTSD are more likely to experience painful TMD, AB or SB, whereas type of traumatic event can be of influence [50]. 

The literature associating psychosocial factors, such as stress depression and anxiety, with the onset or aggravation of painful TMDs is abundant [1,6,7,8]. For example, it was reported that TMD patients diagnosed with myofascial pain with referral have a significant Axis II component [40].

Co-occurrence of anxiety, depression and body pain is common and bi-directional [51]. It can be explained by neuroplastic changes in the central nervous system [52] caused by decrease in neurotransmitters such as norepinephrine and 5-hydroxytryptamine [53]. The resulting inflammatory response can in turn aggravate pain and depression [54] by inducing changes in neurotransmitter metabolism, neuroendocrine functions, and neuroplasticity. 

Beside the high levels of stress, depression, anxiety and trauma involved with occupation with sex work, there are also direct physical aspects of the sex workers’ activity, especially that of supplying oral sex services, which may have a direct influence on TMD. 

Muscle pain and sensations related to muscle fatigue are greatly increased following exhausting exercises. They ameliorate once exercise is ended but increase again 12–24 h later and can last 48 h or longer before diminishing. The pain is often called delayed-onset muscle soreness (DOMS). One explanation for DOMS is that exercise causes micro-tears in the muscle that result in inflammation [55]. In order for the enhanced sensations to be apparent, the sensory receptors encoding muscle pain and fatigue must be sensitized [19]. For example, sensitization of nociceptive pain pathways in the central nervous system due to prolonged nociceptive stimuli from myofascial trigger points seems to be responsible for the conversion of episodic to chronic tension-type headache [56]. Prolonged overuse of the masticatory perioral and tongue muscles and the need for prolonged mouth opening in a sustained muscle position, exercised by sex workers, may cause masticatory muscle fatigue and pain which can explain the difference between groups in the amount of maximal assisted opening. 

An additional diagnosis, more prominent among sex workers as compared to the general population, was Disc displacement with reduction (DDwR). This is a relatively common phenomenon occurring in about 33% of the general population aged 18–55 [57]. The etiological factors include direct external trauma such as a blow to the face, fracture, prolonged opening and/or microtrauma including small force from sustained, repeated load to the masticatory muscles related to mandibular abnormal posture, or parafunctional habits which may act as initiating or perpetuating factors [58,59,60,61]. In the present study, subjects with recent trauma to the face were initially excluded from the analyses. The difference between groups (twice higher incidence of DDwR among sex workers, as compared to the general population) may be an additional consequence of prolonged mouth opening or/and micro-trauma caused by intra-oral soft tissue biting (14.2% among sex workers compared to only 2.4% in the general population). Oral parafunctions are non-physiological oral functions that do not serve any functional purpose [62]. When the activity exceeds physiologic tolerance breakdown may occur leading to the development of orofacial pain and TMD [63,64,65].

The prevalence of probable SB and probable AB in both groups (ranging from 20–30%) is in line with previous studies performed in Israel on the general population [66,67]. It was suggested that SB should be evaluated in its continuum spectrum, rather than using a simplified dichotomous approach of presence/absence [68]. In the present study, no differences between groups were observed in the presence of clinical signs supporting SB and/or AB behavior. This suggests that sex work does not increase the odds of SB and/or AB among this population.

This is one of the first studies addressing TMD among sex workers. Most studies referring to this population address issues such as venereal diseases, HIV, trauma and psychological/psychiatric aspects. Yet it is obvious that sex workers experience more extensive health problems. Oral health, TMD and oral parafunctions are some of the additional health issues that should be addressed and explored in this population.

No study is without limitations. In the present study the group of sex workers was relatively small and the number of subjects in the two examined groups (SW and GP) was imbalanced. Furthermore, DC/TMD Axis II was not included in the study, so that results could not confirm the suggested TMD/psychosocial association. It is noteworthy that dealing with sex workers is very sensitive and should be carried out with extreme emotional sensitivity. Further studies with a larger number of SW subjects on both Axis I and Axis II of the DC/TMD are recommended to further address this issue.

## 5. Conclusions

To the best of our knowledge this is one of the first studies addressing TMD among sex workers. The finding that odds of painful TMD among sex workers are twice to over five times higher than in the general population is alarming. It is essential to develop proper health policies and tailored interventions to assess and control the etiological factors that may cause, perpetuate and/or aggravate TMD among this population.

## Figures and Tables

**Table 1 jcm-11-06622-t001:** Oral parafunctions.

Parafunction	General Population (%)	Sex Workers (%)
Nail biting	16.3	20.0
Chewing gum	3.5	2.9
Holding hard objects in the mouth	1.7	2.9
**Biting soft oral tissues ***	**2.4**	**14.3**
Clenching	24.0	22.9
Do not know	2.4	0

* Significant difference marked in bold, *p* < 0.05, Chi square.

**Table 2 jcm-11-06622-t002:** Opening, lateral and protrusive movements.

Movement	General Population (mm)	Sex Workers (mm)	*p* (*t* Test)
Maximum unassisted opening	50.39 ± 6.90	52.90 ± 7.53	NS
**Maximum assisted opening ***	**52.30 ± 52.30**	**55.24 ± 7.06**	**0.028**
Right Lateral protrusion	9.22 ± 3.00	8.93 3.56	NS
Left Lateral protrusion	9.31 ± 2.91	9.23 ± 3.30	NS
Protrusion	5.64 ± 2.28	5.86 ± 0.49	NS

* *p* < 0.05, *t*-test.

**Table 3 jcm-11-06622-t003:** TMD diagnoses.

TMD Diagnosis *	General Population (%)	Sex Workers (%)	*p* **
**TMJ Disorders: DDwR**	**14.6**	**30.3**	**0.042**
DwRwIL	3.1	2.9	NS
DDwoRwLO	0.0	0	NS
DDwoRwoLO	0.3	2.9	NS
Degenerative Joint Disease	0.0	2.9	NS
Subluxation	0.7	0	NS
**Pain Disorders: Myalgia**	**16.3**	**28.6**	**0.054**
**Myofascial pain with referral**	**6.3**	**20.0**	**0.008**
**Right Arthralgia**	**3.1**	**14.3**	**0.008**
**Left Arthralgia**	**3.8**	**17.1**	**0.004**
**Headache attributed to TMD**	**8.3**	**20.0**	**0.026**

* Diagnoses as follows: DDwR-disc displacement with reduction; DDwRwIL- disc displacement with reduction with intermittent locking; DDwoRwLO- disc displacement without reduction with limited opening; DDwoRwoLO- disc displacement without reduction without limited opening. ** Significant differences marked in bold, Chi square.

**Table 4 jcm-11-06622-t004:** Risk estimates of TMD diagnoses for the two groups.

TMD Diagnosis	Odds Ratio (GP/SW) *	95% C.I	
		Upper	Lower
**DDwR ****	2.545	1.129	5.738
**Myalgia**	2.263	1.006	5.090
**Myofascial pain with referral**	4.060	1.547	10.653
**Right Arthralgia**	5.597	1.750	17.899
**Left Arthralgia**	5.664	1.937	16.564
**Headache attributed to TMD**	2.998	1.175	7.651

* Sex workers versus general population ** DDwR—disc displacement with reduction.

**Table 5 jcm-11-06622-t005:** Probable SB, probable AB and clinical signs supporting their presence.

Variable	General Population (%)	Sex Workers (%)
Linea Alba	85.4	77.1
Abfractions	2.4	0
Indentation	74.6	69.7
Dental Wear	78.1	74.3
Probable AB	31.4	28.5
Probable SB	28.6	20.8

## Data Availability

Datasets generated and/or analyzed during the current study are available from the corresponding author upon reasonable request.

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
