# Peer review of "Temporomandibular Disorders and Bruxism among Sex Workers—A Cross Sectional Study"

_jcm, 2022, doi:10.3390/jcm11226622_

Round 1

Reviewer 1 Report

The aim of this study is very interesting, and it should be extended to evaluate health issues concerning sexual workers. The results of this study are of public health interest. The study is well written, the methods and results are well presented.

I only recommend minor english grammatic checking

Author Response

Dear reviewer,

Thank you for your time and thorough report.

As to your request, grammatic checking was performed.

Reviewer 2 Report

Dear Authors,

The cross sectional study you proposed accurately presents the current situation of a specific social group in your country. Admitting the study group was not so numerous, it still highlights an so far unpopular problem, witch is a vital contribution within the area of epidemiological studies.The study is of scientific interest and in line with the scope of the journal.
However, there are some minor issues that should be addressed.

1. Please remove the informations about the population structure from Results section (lines 149-156) and place it within the Population paragraph.

2. The sum up of the analyzed variables (line 137) needs a separate paragraph, should not be a part of "3. Oral parafunctional activity" paragraph.

3. Please place in order the Results section Tables in line with the list of analyzed variables as you suggested in line 137: 1) TMD pain, 2)Parafunction, 3)TMD diagnosis, etc. - it will make your manuscript mor clear and organized

4. The aim of the study was to investigate the TMD and bruxism prevalence within the sex workers group. Those variables were put together, as I understand, due to the hypothesis that sex work might increase the risk of those disorders with increased stress (for various reasons) as a main reason for that risk. In my opinion the Introduction section should include a better description how increased stress can contribute to TMD and bruxism (around 49 line?). Please mention other possible causes of TMD and bruxism specific for this social group in Introduction (e.g. excessive masticatory muscles fatigue).

5. line 35 - what do you mean by exogenous hormones? 

6. line 64 - I think it might be useful and interesting to expand information what substances and in what way can contribute to TMD/bruxism (alcohol, drugs).

7. line 163 - please make clear for the reader why those variables are vital in the light of the results

8. line 173 and 176 - please keep the descriptions consistent for better clarity - if you decide to describe the GP vs SW in line 173, please correct the line 176 informations accordingly.

9. line 243 - please explain in greater detail the process of repetitive excessive muscle fatigue contribution to development of chronic myalgia and TMD (explain myofibrotic contractures, local myalgia, myofascial pain, etc.)

10. line 263 - the paragraph is just a repetition of the results, please reflect what are the possible reasons for such results

11. Please correct numerous spelling and grammar mistakes.

Thank you for your collaboration and good luck!

Zuzanna Nowak

Author Response

Dear reviewer,

Thank you for your time your thorough report and your valuable comments which no doubt helped improve the manuscript.

Enclosed please find our point-by-point answers to your suggestions/comments.

  1. Please remove the information about the population structure from Results section (lines 149-156) and place it within the Population paragraph.

Response:  Information about the population was removed and placed as requested in the population paragraph highlighted yellow.

  1. The sum up of the analyzed variables (line 137) needs a separate paragraph, should not be a part of "3. Oral parafunctional activity" paragraph.

 Response:  Corrected.

  1. Please place in order the Results section Tables in line with the list of analyzed variables as you suggested in line 137: 1) TMD pain, 2) Parafunction, 3) TMD diagnosis, etc. - it will make your manuscript mor clear and organized

Response:  tables were replaced in the order of the report of analyzed variables please refer to result section.

  1. The aim of the study was to investigate the TMD and bruxism prevalence within the sex workers group. Those variables were put together, as I understand, due to the hypothesis that sex work might increase the risk of those disorders with increased stress (for various reasons) as a main reason for that risk. In my opinion the Introduction section should include a better description how increased stress can contribute to TMD and bruxism (around 49 line?). Please mention other possible causes of TMD and bruxism specific for this social group in Introduction (e.g. excessive masticatory muscles fatigue).

 Response: The topic was expanded as requested in the introduction section, highlighted in yellow to your convenience.

  1. line 35 - what do you mean by exogenous hormones?

 Response:  The phrase was removed.

  1. line 64 - I think it might be useful and interesting to expand information what substances and in what way can contribute to TMD/bruxism (alcohol, drugs).

 Response: The topic was expanded as requested at the end of the introduction section. highlighted in yellow to your convenience.

  1. line 163 - please make clear for the reader why those variables are vital in the light of the results

 Response: An explanation was added. Please refer to the 1st paragraph in the result section, highlighted in yellow to your convenience.

  1. line 173 and 176 - please keep the descriptions consistent for better clarity - if you decide to describe the GP vs SW in line 173, please correct the line 176 information accordingly.

Response:  Corrected

  1. line 243 - please explain in greater detail the process of repetitive excessive muscle fatigue contribution to development of chronic myalgia and TMD (explain myofibrotic contractures, local myalgia, myofascial pain, etc.).

 Response: The topic was expanded as requested. Two paragraphs were added to the discussion section, highlighted in yellow to your convenience.

  1. line 263 - the paragraph is just a repetition of the results, please reflect what are the possible reasons for such results

Response: The topic was expanded as requested. Please refer to the end of the discussion section, highlighted in yellow to your convenience.

  1. Please correct numerous spelling and grammar mistakes.

Response: as to your request, grammatic checking was performed.

Dear reviewer,

Thank you for your time your thorough report and your valuable comments which no doubt helped improve the manuscript.

Enclosed please find our point-by-point answers to your suggestions/comments.

  1. Please remove the information about the population structure from Results section (lines 149-156) and place it within the Population paragraph.

Response:  Information about the population was removed and placed as requested in the population paragraph highlighted yellow.

  1. The sum up of the analyzed variables (line 137) needs a separate paragraph, should not be a part of "3. Oral parafunctional activity" paragraph.

 Response:  Corrected.

  1. Please place in order the Results section Tables in line with the list of analyzed variables as you suggested in line 137: 1) TMD pain, 2) Parafunction, 3) TMD diagnosis, etc. - it will make your manuscript mor clear and organized

Response:  tables were replaced in the order of the report of analyzed variables please refer to result section.

  1. The aim of the study was to investigate the TMD and bruxism prevalence within the sex workers group. Those variables were put together, as I understand, due to the hypothesis that sex work might increase the risk of those disorders with increased stress (for various reasons) as a main reason for that risk. In my opinion the Introduction section should include a better description how increased stress can contribute to TMD and bruxism (around 49 line?). Please mention other possible causes of TMD and bruxism specific for this social group in Introduction (e.g. excessive masticatory muscles fatigue).

 Response: The topic was expanded as requested in the introduction section, highlighted in yellow to your convenience.

  1. line 35 - what do you mean by exogenous hormones?

 Response:  The phrase was removed.

  1. line 64 - I think it might be useful and interesting to expand information what substances and in what way can contribute to TMD/bruxism (alcohol, drugs).

 Response: The topic was expanded as requested at the end of the introduction section. highlighted in yellow to your convenience.

  1. line 163 - please make clear for the reader why those variables are vital in the light of the results

 Response: An explanation was added. Please refer to the 1st paragraph in the result section, highlighted in yellow to your convenience.

  1. line 173 and 176 - please keep the descriptions consistent for better clarity - if you decide to describe the GP vs SW in line 173, please correct the line 176 information accordingly.

Response:  Corrected

  1. line 243 - please explain in greater detail the process of repetitive excessive muscle fatigue contribution to development of chronic myalgia and TMD (explain myofibrotic contractures, local myalgia, myofascial pain, etc.).

 Response: The topic was expanded as requested. Two paragraphs were added to the discussion section, highlighted in yellow to your convenience.

  1. line 263 - the paragraph is just a repetition of the results, please reflect what are the possible reasons for such results

Response: The topic was expanded as requested. Please refer to the end of the discussion section, highlighted in yellow to your convenience.

  1. Please correct numerous spelling and grammar mistakes.

Response: as to your request, grammatic checking was performed.

Dear reviewer,

Thank you for your time your thorough report and your valuable comments which no doubt helped improve the manuscript.

Enclosed please find our point-by-point answers to your suggestions/comments.

  1. Please remove the information about the population structure from Results section (lines 149-156) and place it within the Population paragraph.

Response:  Information about the population was removed and placed as requested in the population paragraph highlighted yellow.

  1. The sum up of the analyzed variables (line 137) needs a separate paragraph, should not be a part of "3. Oral parafunctional activity" paragraph.

 Response:  Corrected.

  1. Please place in order the Results section Tables in line with the list of analyzed variables as you suggested in line 137: 1) TMD pain, 2) Parafunction, 3) TMD diagnosis, etc. - it will make your manuscript mor clear and organized

Response:  tables were replaced in the order of the report of analyzed variables please refer to result section.

  1. The aim of the study was to investigate the TMD and bruxism prevalence within the sex workers group. Those variables were put together, as I understand, due to the hypothesis that sex work might increase the risk of those disorders with increased stress (for various reasons) as a main reason for that risk. In my opinion the Introduction section should include a better description how increased stress can contribute to TMD and bruxism (around 49 line?). Please mention other possible causes of TMD and bruxism specific for this social group in Introduction (e.g. excessive masticatory muscles fatigue).

 Response: The topic was expanded as requested in the introduction section, highlighted in yellow to your convenience.

  1. line 35 - what do you mean by exogenous hormones?

 Response:  The phrase was removed.

  1. line 64 - I think it might be useful and interesting to expand information what substances and in what way can contribute to TMD/bruxism (alcohol, drugs).

 Response: The topic was expanded as requested at the end of the introduction section. highlighted in yellow to your convenience.

  1. line 163 - please make clear for the reader why those variables are vital in the light of the results

 Response: An explanation was added. Please refer to the 1st paragraph in the result section, highlighted in yellow to your convenience.

  1. line 173 and 176 - please keep the descriptions consistent for better clarity - if you decide to describe the GP vs SW in line 173, please correct the line 176 information accordingly.

Response:  Corrected

  1. line 243 - please explain in greater detail the process of repetitive excessive muscle fatigue contribution to development of chronic myalgia and TMD (explain myofibrotic contractures, local myalgia, myofascial pain, etc.).

 Response: The topic was expanded as requested. Two paragraphs were added to the discussion section, highlighted in yellow to your convenience.

  1. line 263 - the paragraph is just a repetition of the results, please reflect what are the possible reasons for such results

Response: The topic was expanded as requested. Please refer to the end of the discussion section, highlighted in yellow to your convenience.

  1. Please correct numerous spelling and grammar mistakes.

Response: as to your request, grammatic checking was performed.

Dear reviewer,

Thank you for your time your thorough report and your valuable comments which no doubt helped improve the manuscript.

Enclosed please find our point-by-point answers to your suggestions/comments.

  1. Please remove the information about the population structure from Results section (lines 149-156) and place it within the Population paragraph.

Response:  Information about the population was removed and placed as requested in the population paragraph highlighted yellow.

  1. The sum up of the analyzed variables (line 137) needs a separate paragraph, should not be a part of "3. Oral parafunctional activity" paragraph.

 Response:  Corrected.

  1. Please place in order the Results section Tables in line with the list of analyzed variables as you suggested in line 137: 1) TMD pain, 2) Parafunction, 3) TMD diagnosis, etc. - it will make your manuscript mor clear and organized

Response:  tables were replaced in the order of the report of analyzed variables please refer to result section.

  1. The aim of the study was to investigate the TMD and bruxism prevalence within the sex workers group. Those variables were put together, as I understand, due to the hypothesis that sex work might increase the risk of those disorders with increased stress (for various reasons) as a main reason for that risk. In my opinion the Introduction section should include a better description how increased stress can contribute to TMD and bruxism (around 49 line?). Please mention other possible causes of TMD and bruxism specific for this social group in Introduction (e.g. excessive masticatory muscles fatigue).

 Response: The topic was expanded as requested in the introduction section, highlighted in yellow to your convenience.

  1. line 35 - what do you mean by exogenous hormones?

 Response:  The phrase was removed.

  1. line 64 - I think it might be useful and interesting to expand information what substances and in what way can contribute to TMD/bruxism (alcohol, drugs).

 Response: The topic was expanded as requested at the end of the introduction section. highlighted in yellow to your convenience.

  1. line 163 - please make clear for the reader why those variables are vital in the light of the results

 Response: An explanation was added. Please refer to the 1st paragraph in the result section, highlighted in yellow to your convenience.

  1. line 173 and 176 - please keep the descriptions consistent for better clarity - if you decide to describe the GP vs SW in line 173, please correct the line 176 information accordingly.

Response:  Corrected

  1. line 243 - please explain in greater detail the process of repetitive excessive muscle fatigue contribution to development of chronic myalgia and TMD (explain myofibrotic contractures, local myalgia, myofascial pain, etc.).

 Response: The topic was expanded as requested. Two paragraphs were added to the discussion section, highlighted in yellow to your convenience.

  1. line 263 - the paragraph is just a repetition of the results, please reflect what are the possible reasons for such results

Response: The topic was expanded as requested. Please refer to the end of the discussion section, highlighted in yellow to your convenience.

  1. Please correct numerous spelling and grammar mistakes.

Response: as to your request, grammatic checking was performed.

Dear reviewer,

Thank you for your time your thorough report and your valuable comments which no doubt helped improve the manuscript.

Enclosed please find our point-by-point answers to your suggestions/comments.

  1. Please remove the information about the population structure from Results section (lines 149-156) and place it within the Population paragraph.

Response:  Information about the population was removed and placed as requested in the population paragraph highlighted yellow.

  1. The sum up of the analyzed variables (line 137) needs a separate paragraph, should not be a part of "3. Oral parafunctional activity" paragraph.

 Response:  Corrected.

  1. Please place in order the Results section Tables in line with the list of analyzed variables as you suggested in line 137: 1) TMD pain, 2) Parafunction, 3) TMD diagnosis, etc. - it will make your manuscript mor clear and organized

Response:  tables were replaced in the order of the report of analyzed variables please refer to result section.

  1. The aim of the study was to investigate the TMD and bruxism prevalence within the sex workers group. Those variables were put together, as I understand, due to the hypothesis that sex work might increase the risk of those disorders with increased stress (for various reasons) as a main reason for that risk. In my opinion the Introduction section should include a better description how increased stress can contribute to TMD and bruxism (around 49 line?). Please mention other possible causes of TMD and bruxism specific for this social group in Introduction (e.g. excessive masticatory muscles fatigue).

 Response: The topic was expanded as requested in the introduction section, highlighted in yellow to your convenience.

  1. line 35 - what do you mean by exogenous hormones?

 Response:  The phrase was removed.

  1. line 64 - I think it might be useful and interesting to expand information what substances and in what way can contribute to TMD/bruxism (alcohol, drugs).

 Response: The topic was expanded as requested at the end of the introduction section. highlighted in yellow to your convenience.

  1. line 163 - please make clear for the reader why those variables are vital in the light of the results

 Response: An explanation was added. Please refer to the 1st paragraph in the result section, highlighted in yellow to your convenience.

  1. line 173 and 176 - please keep the descriptions consistent for better clarity - if you decide to describe the GP vs SW in line 173, please correct the line 176 information accordingly.

Response:  Corrected

  1. line 243 - please explain in greater detail the process of repetitive excessive muscle fatigue contribution to development of chronic myalgia and TMD (explain myofibrotic contractures, local myalgia, myofascial pain, etc.).

 Response: The topic was expanded as requested. Two paragraphs were added to the discussion section, highlighted in yellow to your convenience.

  1. line 263 - the paragraph is just a repetition of the results, please reflect what are the possible reasons for such results

Response: The topic was expanded as requested. Please refer to the end of the discussion section, highlighted in yellow to your convenience.

  1. Please correct numerous spelling and grammar mistakes.

Response: as to your request, grammatic checking was performed.

Dear reviewer,

Thank you for your time your thorough report and your valuable comments which no doubt helped improve the manuscript.

Enclosed please find our point-by-point answers to your suggestions/comments.

  1. Please remove the information about the population structure from Results section (lines 149-156) and place it within the Population paragraph.

Response:  Information about the population was removed and placed as requested in the population paragraph highlighted yellow.

  1. The sum up of the analyzed variables (line 137) needs a separate paragraph, should not be a part of "3. Oral parafunctional activity" paragraph.

 Response:  Corrected.

  1. Please place in order the Results section Tables in line with the list of analyzed variables as you suggested in line 137: 1) TMD pain, 2) Parafunction, 3) TMD diagnosis, etc. - it will make your manuscript mor clear and organized

Response:  tables were replaced in the order of the report of analyzed variables please refer to result section.

  1. The aim of the study was to investigate the TMD and bruxism prevalence within the sex workers group. Those variables were put together, as I understand, due to the hypothesis that sex work might increase the risk of those disorders with increased stress (for various reasons) as a main reason for that risk. In my opinion the Introduction section should include a better description how increased stress can contribute to TMD and bruxism (around 49 line?). Please mention other possible causes of TMD and bruxism specific for this social group in Introduction (e.g. excessive masticatory muscles fatigue).

 Response: The topic was expanded as requested in the introduction section, highlighted in yellow to your convenience.

  1. line 35 - what do you mean by exogenous hormones?

 Response:  The phrase was removed.

  1. line 64 - I think it might be useful and interesting to expand information what substances and in what way can contribute to TMD/bruxism (alcohol, drugs).

 Response: The topic was expanded as requested at the end of the introduction section. highlighted in yellow to your convenience.

  1. line 163 - please make clear for the reader why those variables are vital in the light of the results

 Response: An explanation was added. Please refer to the 1st paragraph in the result section, highlighted in yellow to your convenience.

  1. line 173 and 176 - please keep the descriptions consistent for better clarity - if you decide to describe the GP vs SW in line 173, please correct the line 176 information accordingly.

Response:  Corrected

  1. line 243 - please explain in greater detail the process of repetitive excessive muscle fatigue contribution to development of chronic myalgia and TMD (explain myofibrotic contractures, local myalgia, myofascial pain, etc.).

 Response: The topic was expanded as requested. Two paragraphs were added to the discussion section, highlighted in yellow to your convenience.

  1. line 263 - the paragraph is just a repetition of the results, please reflect what are the possible reasons for such results

Response: The topic was expanded as requested. Please refer to the end of the discussion section, highlighted in yellow to your convenience.

  1. Please correct numerous spelling and grammar mistakes.

Response: as to your request, grammatic checking was performed.

Reviewer 3 Report

Dear Authors,

thank you for submitting.

Your paper is very well-designed and "The tooth fairy" is a very important social project.

Even if the target of this paper is very unusual - sex workers, the reserach design is flawless (the best that I found as MDPI reviewer). Moreover this paper is the result of a social project.   The aim of the is to evaluate the extent of TMD and bruxism among sex 65 workers in Israel as compared to the general population. The way in which the Authors developed the research design denotes a high degree of knowing of TMD and bruxism.  The study shows that health problems of health workers are not only venereal diseases, HIV and mental disorders but also oral pathology, TMD and parafunctions.  The paper is well written, very fine English, clear and easy to read. The conclusions are consistent with the evidence and arguments presented.

Author Response

Dear reviewer,

Thank you for your time, thorough report and your kind remarques.

Reviewer 4 Report

The manuscript is well written and focus on an interesting topic. Please find my minor comments below:

 Materials and methods

Lines 71-75 – this fragment should be deleted or moved to discussion or introduction.

Results

Table 4, 5 – please add explanations of abbreviations under the table

Author Response

Dear reviewer,

Thank you for your time, thorough report, and comments.

Enclosed please find our point-by-point answers to your comments/sugestions

  • Materials and methods

Lines 71-75 – this fragment should be deleted or moved to discussion or introduction.

Response: as to your suggestion this fragment was moved to the end of the introduction section.

To your convenience highlighted in green.

  • Results

Table 4, 5 – please add explanations of abbreviations under the table

Response: abbreviations were added and highlighted in green.

Dear reviewer,

Thank you for your time, thorough report, and comments.

Enclosed please find our point-by-point answers to your comments/sugestions

  • Materials and methods

Lines 71-75 – this fragment should be deleted or moved to discussion or introduction.

Response: as to your suggestion this fragment was moved to the end of the introduction section.

To your convenience highlighted in green.

  • Results

Table 4, 5 – please add explanations of abbreviations under the table

Response: abbreviations were added and highlighted in green.

Dear reviewer,

Thank you for your time, thorough report, and comments.

Enclosed please find our point-by-point answers to your comments/sugestions

  • Materials and methods

Lines 71-75 – this fragment should be deleted or moved to discussion or introduction.

Response: as to your suggestion this fragment was moved to the end of the introduction section.

To your convenience highlighted in green.

  • Results

Table 4, 5 – please add explanations of abbreviations under the table

Response: abbreviations were added and highlighted in green.

Dear reviewer,

Thank you for your time, thorough report, and comments.

Enclosed please find our point-by-point answers to your comments/sugestions

  • Materials and methods

Lines 71-75 – this fragment should be deleted or moved to discussion or introduction.

Response: as to your suggestion this fragment was moved to the end of the introduction section.

To your convenience highlighted in green.

  • Results

Table 4, 5 – please add explanations of abbreviations under the table

Response: abbreviations were added and highlighted in green.

Dear reviewer,

Thank you for your time, thorough report, and comments.

Enclosed please find our point-by-point answers to your comments/sugestions

  • Materials and methods

Lines 71-75 – this fragment should be deleted or moved to discussion or introduction.

Response: as to your suggestion this fragment was moved to the end of the introduction section.

To your convenience highlighted in green.

  • Results

Table 4, 5 – please add explanations of abbreviations under the table

Response: abbreviations were added and highlighted in green.

Dear reviewer,

Thank you for your time, thorough report, and comments.

Enclosed please find our point-by-point answers to your comments/sugestions

  • Materials and methods

Lines 71-75 – this fragment should be deleted or moved to discussion or introduction.

Response: as to your suggestion this fragment was moved to the end of the introduction section.

To your convenience highlighted in green.

  • Results

Table 4, 5 – please add explanations of abbreviations under the table

Response: abbreviations were added and highlighted in green.
